DISCOVERY REPORT

# The dominantly expressed class II molecule from a resistant MHC haplotype presents only a few Marek's disease virus peptides by using an unprecedented binding motif

**Samer Halabi**[1,2], **Michael Ghosh**[3], **Stefan Stevanović**[3], **Hans-Georg Rammensee**[3], **Luca D. Bertzbach**[4], **Benedikt B. Kaufer**[4], **Martin C. Moncrieffe**[5], **Bernd Kaspers**[6], **Sonja Härtle**[6]*, **Jim Kaufman**[1,2,7]*

1 University of Cambridge, Department of Pathology, Cambridge, United Kingdom, 2 University of Edinburgh, Institute for Immunology and Infection Research, Edinburgh, United Kingdom, 3 University of Tübingen, Department of Immunology, Institute of Cell Biology, Tübingen, Germany, 4 Freie Universität Berlin, Institut für Virologie, Berlin, Germany, 5 University of Cambridge, Department of Biochemistry, Cambridge, United Kingdom, 6 Ludwig Maximillians University, Veterinary Faculty, Planegg, Germany, 7 University of Cambridge, Department of Veterinary Medicine, Cambridge, United Kingdom

* sonja.haertle@lmu.de (SH); jim.kaufman@ed.ac.uk (JK)

## Abstract

Viral diseases pose major threats to humans and other animals, including the billions of chickens that are an important food source as well as a public health concern due to zoonotic pathogens. Unlike humans and other typical mammals, the major histocompatibility complex (MHC) of chickens can confer decisive resistance or susceptibility to many viral diseases. An iconic example is Marek's disease, caused by an oncogenic herpesvirus with over 100 genes. Classical MHC class I and class II molecules present antigenic peptides to T lymphocytes, and it has been hard to understand how such MHC molecules could be involved in susceptibility to Marek's disease, given the potential number of peptides from over 100 genes. We used a new in vitro infection system and immunopeptidomics to determine peptide motifs for the 2 class II molecules expressed by the MHC haplotype B2, which is known to confer resistance to Marek's disease. Surprisingly, we found that the vast majority of viral peptide epitopes presented by chicken class II molecules arise from only 4 viral genes, nearly all having the peptide motif for BL2*02, the dominantly expressed class II molecule in chickens. We expressed BL2*02 linked to several Marek's disease virus (MDV) peptides and determined one X-ray crystal structure, showing how a single small amino acid in the binding site causes a crinkle in the peptide, leading to a core binding peptide of 10 amino acids, compared to the 9 amino acids in all other reported class II molecules. The limited number of potential T cell epitopes from such a complex virus can explain the differential MHC-determined resistance to MDV, but raises questions of mechanism and opportunities for vaccine targets in this important food species, as well as providing a basis for understanding class II molecules in other species including humans.

**Data Availability Statement:** All raw mass spectrometry proteomics data have been deposited to the ProteomeXchange Consortium via the PRIDE partner repository with the dataset identifier PXD023954. All analysed data are within the paper and its Supporting Information files.

**Funding:** This work was supported by the German Cancer Consortium (DKTK) and the NMI Natural and Medical Sciences Institute at the University of Tübingen to H-GR and SS, the Volkswagen Foundation Lichtenberg grant A112662 awarded to BBK, the BMBF FugatoPlus project AvImmun and ERA Net ANIHWA-MADISUP to SHaertle, and an Investigator Award (110106/Z/15/Z) from the Wellcome Trust awarded to JK. The funders had no role in the study design, data collection, decision to publish, or preparation of the manuscript.

**Competing interests:** The authors have declared that no competing interests exist.

**Abbreviations:** BAC, bacterial artificial chromosome; CEC, chicken embryo cell; FDR, false discovery rate; FRET, Fluorescence Resonance Energy Transfer; gE, glycoprotein E; GFP, green fluorescent protein; gH, glycoprotein H; gI, glycoprotein I; H-bonds, hydrogen bonds; HSV-1, herpes simplex virus 1; IMDM, Iscove's modified Dulbecco's medium; MDV, Marek's disease virus; MEM, minimal essential medium; MHC, major histocompatibility complex; NK, natural killer; PDB, Protein Database; PE, phycoerythrin; SPF, specific-pathogen-free; TEV, tobacco etch virus; TFA, trifluoroacetic acid; UL43, unique long gene 43.

## Introduction

The ongoing global pandemic of a coronavirus among humans highlights the enormous challenge of viral disease and the importance of the appropriate immune responses [1–3]. The classical class I and class II molecules of the major histocompatibility complex (MHC) play crucial roles in resistance to infection and response to vaccines, binding peptides for presentation to thymus-derived (T) lymphocytes of the adaptive immune system as well as natural killer (NK) cells of the innate immune system [4,5]. The importance of the classical MHC molecules is underscored by their high level of allelic polymorphism, which is mostly driven by molecular arms races with infectious pathogens [5]. However, the MHC of humans and typical mammals is an enormous and complex genetic region encoding a wide variety of molecules, with multigene families encoding class I and class II molecules leading to strong genetic associations with autoimmune disease but relatively weak associations with infectious diseases [4].

In contrast to humans and other typical mammals, the level of resistance to many infectious pathogens in chickens can be strongly determined by the MHC (that is, the BF-BL region of the B locus), at least in part because the chicken MHC is much simpler, with single dominantly expressed class I and class II loci [6,7]. As a result of this simplicity, the phenotypes are much clearer: either the dominantly expressed MHC allele finds a protective peptide to confer resistance or not, leading to strong genetic associations that are easier to discover and dissect [7]. Moreover, the scale of viral challenges in the poultry industry has been clear for decades, with many tens of billions of chickens each year beset by a wide variety of poultry viruses [6], including the first described coronavirus [8]. On top of the economic importance, zoonotic pathogens (including avian influenza) have been a continuing concern for public health [9,10]. Despite enormous efforts in biosecurity, vaccination, and genetic breeding, condemnation and slaughter of huge numbers of infected chickens are relatively frequent [11].

Marek's disease (MD), caused by an oncogenic herpesvirus, was the first reported disease for which the chicken MHC is known to determine resistance and susceptibility and remains an enormous burden for the poultry industry, with continuing outbreaks despite routine vaccination [12–16]. Indeed, current vaccines control disease but not transmission, leading to selection of more virulent strains, which in turn have required more efficacious vaccines [17,18]. The virus responsible for MD (MDV), in common with other herpesviruses, has a relatively large genome with over 100 genes and a complex life cycle, so it is possible that many genes contribute to resistance at different stages of infection, tumor growth, and transmission [14,16]. Several polymorphic genes located in the MHC have been proposed as candidates to determine MD resistance, including the dominantly expressed classical class I gene (BF2), an NK receptor gene (B-NK), a gene with some similarities to mammalian butyrophilins (BG1) and the classical class II B genes (BLB1 and BLB2) [19–25]. In comparison to the MHC class I system, very little attention has been focused on chicken class II genes and molecules [7,26].

Mammalian class II molecules have been intensively studied, so many structural and functional features are known in detail [27–30]. The heterodimer is composed of an α and a β chain, each with 2 extracellular domains: membrane proximal immunoglobulin C-like (Ig C) domains (α2 and β2 domains) and membrane distal domains composed of 4 β-strands forming a β-sheet surmounted by predominantly α-helical stretches (α1 and β1 domains). The domain organization, including the features of secondary structure and location of intradomain disulfide bonds, is similar to class I molecules. The membrane distal domains together form a peptide-binding superdomain, with a core sequence from the peptide binding across a groove, the T cell receptor recognizing the top, and the dedicated chaperone DM interacting with key residues on the side for loading of appropriate peptides. A glycosylation site near the end of the α1 domain is likely to be involved in quality control during biosynthesis and peptide

loading (as in class I [31]), while a loop in the β2 domain and part of the α2 domain bind the CD4 coreceptor that contributes to T cell signaling (in the same place as CD8 interacts with class I [30,32]).

As in human HLA-DR molecules, the chicken α-chain (encoded by the BLA gene) is nearly monomorphic, with most residues on the top of the α1 domain identical with HLA-DRA (although there is dimorphic position in the α-helix pointing up towards the T cells, and a four amino acid insertion in one loop of the β-sheet) [33]. Virtually, all of the variation responsible for allelic polymorphism and thus for different peptide-binding specificities in different chicken class II molecules is located in the β1 domain of the β chain, encoded by either the BLB1 or BLB2 gene [34,35].

Differential immune responses by classical MHC molecules have typically been understood as MHC molecules either presenting a peptide that confers an effective immune response or not. However, a major concern is how classical MHC molecules could confer susceptibility to a virus such as MDV, since it is hard to imagine how a peptide conferring protection would not be found among 100 viral molecules [19]. This conundrum is particularly an issue for class II molecules for which the peptide motifs are relatively promiscuous compared to class I molecules [28]. A significant barrier to examining this question has been the low frequency of cells infected with MDV within chickens before tumors arise, even early during infection when MDV replicates mainly in B cells [36].

In this report, we use a novel culture system for bursal B cells [37], followed by determination of the peptides bound to class II molecules by mass spectrometry (so-called immunopeptidomics or MHC ligandomics [38,39]). We identify the peptides bound to class II molecules from the well-characterized MHC haplotype B2 known to confer resistance to MD [13,40], with and without infection by the very virulent MDV strain RB-1B (also known as RB1B [41]) and the live attenuated virus strain CVI988 (widely used as the Rispens vaccine [42]). We then express the chicken class II molecule BL2*02 with several dominant pathogen peptides and determine the structure of one such complex by X-ray crystallography, finding an unusual molecular feature that might help explain the biology of resistance.

## Results

### Peptides from 2 class II molecules were found in bursal B cells from the B2 MHC haplotype

After 1 day in culture with and without infection, ex vivo bursal cells from the M11 chicken line were harvested, MHC molecules were isolated from detergent lysates by affinity chromatography, and the eluted peptides were analyzed by mass spectrometry. In experiments over 3 years, 1 uninfected sample, 2 samples infected with the live attenuated virus strain CVI988, and 3 samples infected with the very virulent MDV strain RB1B were analyzed.

Several thousand peptides were identified for each infected sample, with between 36% and 84% of a particular sample shared with other samples (S1 Fig, S1–S6 Data). Since rare peptides may not be sampled in repeated analyses, the high level of overlap between analyses gives confidence that most abundant peptides detectable by this method were identified.

Overall, the sequences have the features typical of peptides bound by mammalian class II molecules, including an enrichment of proline around position 2 (due to the cleavage preference for aminopeptidases) and a broad distribution of length from 12 to more than 25, with a peak around 16 to 17 amino acids (S2 Fig, S1–S6 Data). Initial analysis by Gibbs clustering identified 2 nonamer (9mer) core motifs in all samples, but subsequent structural evidence (detailed below) led to a reanalysis that showed one of these motifs to be a decamer (10mer) (Fig 1A). The 2 motifs turn out to represent the B2 alleles of the 2 classical class II molecules in

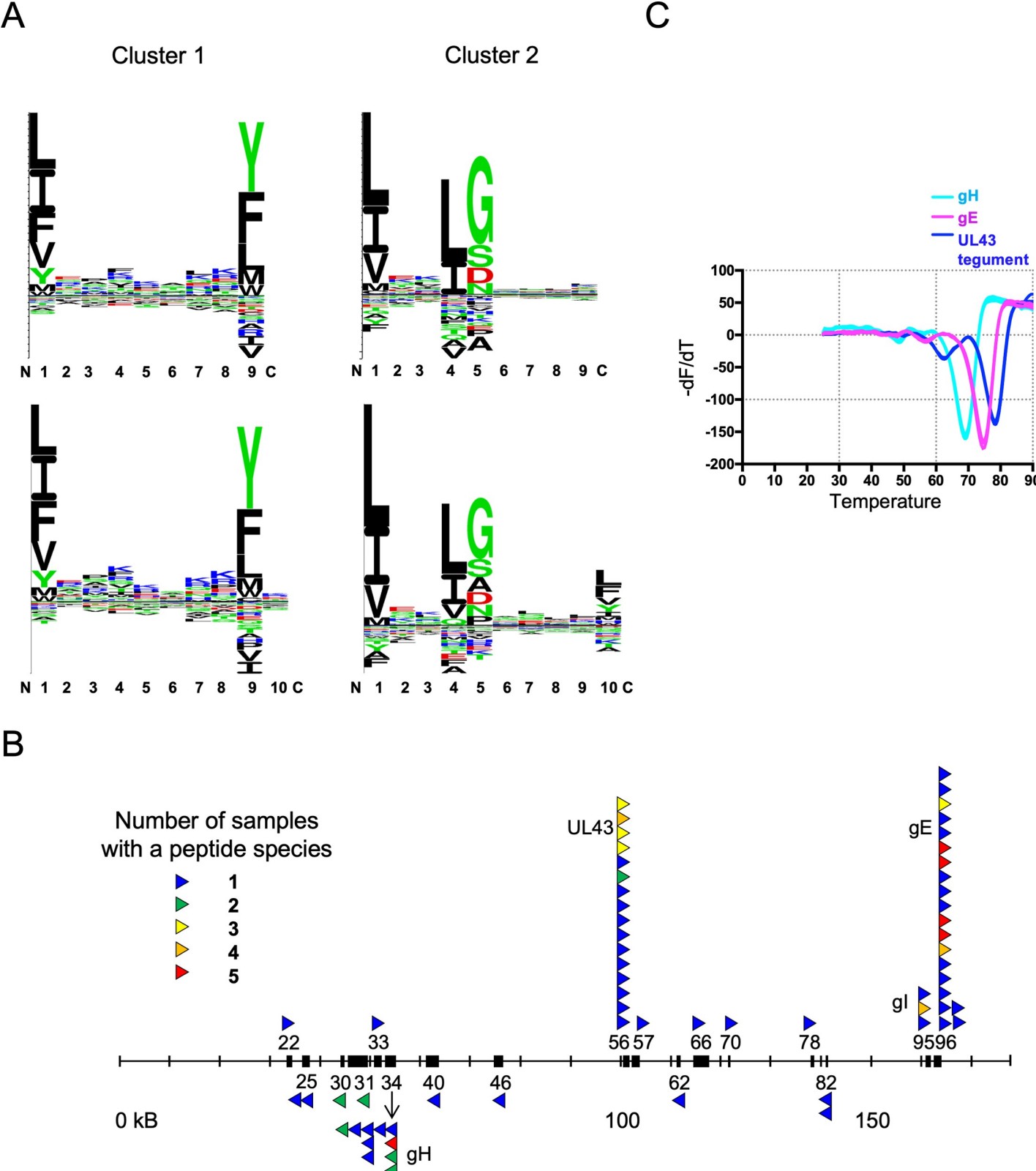

**Fig 1. Analysis of thousands of peptides from infected B2 cells yield 2 peptide motifs but only a few peptides from MDV, most of which are derived from only 4 MDV genes, and with peptides from 3 genes binding BL2*02 molecules with high affinity.** (A) Two peptide motifs are found for class II molecules from the MHC B2

haplotype, one of which is a decamer. Peptide motifs were determined by Gibbs clustering: upper panel with default length set to 9mer, lower panel with default length set to 10mer. These motifs were determined using the CVI988 2020 experiment but are representative of all datasets, alone or taken together. (B) Only 4 MDV proteins give rise to most pathogen peptide epitopes and their various peptide species bound to class II molecules from infected chicken bursal B cells in culture. Thin horizontal line indicates MDV genome of roughly 180 kB, with relevant MDV genes indicated by black boxes labeled with the number of the gene (for instance, MDV022) and with arrowheads indicating class II-bound peptides, above the line for genes and peptides oriented from left to right and below the line for right to left. Each vertical stack of arrowheads indicates a single T cell epitope with multiple peptide species (with the protein name next to the stack), each arrowhead colored according to the number of experimental samples in which the peptide species was found (blue, 1; green, 2; yellow, 3; orange, 4; red, 5). (C) MDV peptides with decamer motifs were expressed with BL2*02 molecules from insect cells, and the expressed peptide-MHC molecules had high thermal stability. Thermal denaturation curves for BL2*02 molecules expressed with GVLF<u>YMPTSHVQQM</u>TF from gH with melting temperature 69˚C, QIES<u>LSLNGVPNIFL</u>STKA from gE with 75˚C, and SSEV<u>LTSIGKPAQFI</u>FA from UL43 tegument protein with 78˚C; the peptide TPSDV<u>IEKELMEKL</u>KKK from gI with a nonamer motif did not express. Motifs in the peptide sequences are underlined. The underlying data for this figure can be found in Fig 2, S1 Table, and S1–S6 Data. gE, glycoprotein E; gH, glycoprotein H; gI, glycoprotein I; MDV, Marek's disease virus; MHC, major histocompatibility complex; UL43, unique long gene 43.

chickens, BL1*02 and BL2*02, encoded by a monomorphic HLA-DRA-like A gene (BLA) and 2 polymorphic classical class II B genes (BLB1 and BLB2). At the RNA level, BLB1 is poorly expressed except in the intestine, while BLB2 is well expressed systemically [26]. However, the peptide species from these ex vivo bursal B cells that contain the BL1 nonamer motif ranged from 37% to 48% of the total containing BL1 and BL2 motifs in different samples (S1–S6 Data).

## Four MDV genes gave rise to most of the pathogen peptides bound to class II molecules from bursal B cells, with most bound to BL2 molecules

Out of the thousands of peptides characterized, the vast majority originated from chicken proteins (S1–S6 Data), but 64 peptides originating from 17 MDV genes were identified (Fig 2, S1 Table). Many of these peptide species and/or epitopes are shared between the live attenuated virus strain CVI988 and the very virulent MDV strain RB1B, with many other peptides unique to CVI988 but none unique to RB1B (Fig 2). The infection rate was much higher for CVI988 than for RB1B in these in vitro cultures, which may have resulted in more class II molecules being loaded with MDV peptides.

Most of the MDV peptides were found only in 1 sample and as only 1 species of peptide (meaning that they are likely to be quite rare). However, overlapping peptides (hereafter called species) corresponding to a single peptide epitope and/or multiple peptide epitopes were found in several infected samples (meaning that they are likely to be abundant) for 4 MDV proteins (Fig 1B, Fig 2, and S1 Table). Five peptide epitopes were found for glycoprotein H (gH, encoded by the MDV034 gene), two of which had multiple species, of which one was found in all 5 samples. The UL43 tegument protein (MDV056) had 1 peptide epitope with 16 species, some of which were found in 2 to 4 samples. For gI (MDV095), 1 peptide epitope was found with 3 species, one of which was found in 4 samples. For gE (MDV096), 2 peptides were found, one of which had 18 species, four of which were found in all 5 samples.

Most of these MDV peptide species had the decamer motif, including those from 3 of the 4 MDV proteins with abundant peptides. From the analysis by Gibbs clustering, 6 peptide species from 4 epitopes fit the nonamer BLB1 motif, 50 species from 13 epitopes fit the decamer BLB2 motif, 1 species from 1 epitope fits both motifs (and thus might actually contain 2 epitopes), and 7 species from 5 epitopes did not fit either motif (Fig 2).

Soluble BL2 molecules from the B2 haplotype were expressed in insect cells using Baculovirus, with the peptide attached via a linker to the BLB2 chain [it should be noted that, in contrast to previous reports [34,43,44], the N-terminal positions of the BLA and BLB chains are likely identical to those of HLA-DR (S3 Fig)]. Confirming the assignments based on motifs, 3 peptide species with decamer motifs linked to BL2*02 molecules were successfully expressed, and the complexes had high thermal stabilities (Fig 1C). However, the one peptide with a

| gene | protein | peptide species | BLB motif | 2018 | RB-1B 2019 | 2020 | CVI988 2019 | 2020 |
|---|---|---|---|---|---|---|---|---|
| MDV022/UL10 | gM | KQRGSQSEDERALTQSRSAE | neither | | | | | 1 |
| MDV025/UL13 | | IEKTFMDLGKAVVFLNVS | 2 | | | | | 1 |
| | | IGDFSLALLNTNSTILK | 2 | | | | 1 | 1 |
| MDV030/UL18 | TRX2 | GTEPDTLSLLSTFKTRFAAVIT | 2 | | 1 | | 1 | |
| MDV031/UL19 | MCP | ISAELVTIGDKLIFLES | 2 | | 1 | | 1 | |
| MDV033/UL21 | | KIGLVLRGGQS | 1 | | | | | 1 |
| MDV034/UL22 | gH | SPANGVLFYMPTSHVQQMTF | 2 | | | | | 1 |
| | | GVLFYMPTSHVQQMTF | 2* | 1 | 1 | 1 | 1 | 1 |
| | | GVLFYMPTSHVQQMT | 2 | 1 | | | | 1 |
| | | GVLFYMPTSHVQQM | 2 | 1 | | | | 1 |
| | | RPVSKLLASNNLIKFLNTG | 2 | | | | | 1 |
| | | ELKNLKPIDVVNPEHRFILT | both | | | | | 1 |
| | | ELKNLKPIDVVNPEHRFI | both | | | | | 1 |
| | | NLKPIDVVNPEHRFILT | both | | | | | 1 |
| | | YDFQIAQTHAQLFI | 2 | | | | | 1 |
| | | TALLVLPISGLGSYVVTRQ | 2 | | | | 1 | 1 |
| MDV040/UL27 | gB | RNQLHELKFYDINKVIE | both | | | | | 1 |
| MDV046/UL32 | | LKHAITKGGTSAE | both | | | | | 1 |
| MDV056/UL43 | | SSEVLTSIGKPAQFIFAL | 2 | | 1 | | 1 | 1 |
| | | SSEVLTSIGKPAQFIFA | 2* | 1 | 1 | | 1 | 1 |
| | | SSEVLTSIGKPAQFIF | 2 | | 1 | | 1 | 1 |
| | | SSEVLTSIGKPAQFI | 2 | 1 | | | 1 | 1 |
| | | SSEVLTSIGKPAQF | 2 | | | | | 1 |
| | | SEVLTSIGKPAQFIFAL | 2 | | | | 1 | 1 |
| | | SEVLTSIGKPAQFIFA | 2 | | | | | 1 |
| | | SEVLTSIGKPAQFIF | 2 | | | | | 1 |
| | | SEVLTSIGKPAQFI | 2 | | | | | 1 |
| | | EVLTSIGKPAQFIFAL | 2 | | | | | 1 |
| | | EVLTSIGKPAQFIFA | 2 | | | | | 1 |
| | | EVLTSIGKPAQFIF | 2 | | | | | 1 |
| | | EVLTSIGKPAQFI | 2 | | | | | 1 |
| | | EVLTSIGKPAQF | 2 | | | | | 1 |
| | | VLTSIGKPAQFIFAL | 2 | | | | | 1 |
| | | VLTSIGKPAQFIFA | 2 | | | | | 1 |
| MDV057/UL44 | gC | GYmQVlmRDHFNRPL | 1 | | | | 1 | |
| MDV062/UL49 | VP22 | TNEELDAFLSRAVIKIT | 1 | | | | | 1 |
| MDV066/UL52 | | VEFCLLNQmASGm | neither | | | | | 1 |
| MDV070/UL55 | | EPVPHIGSITSRSYLLR | both | | | | | 1 |
| MDV078 | vIL-8 | TEIIFALKKNRKV | both | | | | | 1 |
| MDV082 | | SDEPHAPGTTmISGPRTQ | 1 | | | | 1 | |
| | | TTmISGPRTQ | 1 | | | | 1 | |
| MDV095/US7 | gI | TPSDVIEKELMEKLKKKVE | 1 | | | | 1 | |
| | | TPSDVIEKELMEKLKKK | 1** | 1 | | 1 | 1 | 1 |
| | | TPSDVIEKELMEKLKK | 1 | | | | | 1 |
| MDV096/US8 | gE | IKQIESLSLNGVPNIFLSTKA | 2 | | | | | 1 |
| | | KQIESLSLNGVPNIFLSTKA | 2 | | | | | 1 |
| | | QIESLSLNGVPNIFLSTKASNK | 2 | | | 1 | 1 | 1 |
| | | QIESLSLNGVPNIFLSTKASN | 2 | | | | | 1 |
| | | QIESLSLNGVPNIFLSTKAS | 2 | | | | | 1 |
| | | QIESLSLNGVPNIFLSTKA | 2* | 1 | 1 | 1 | 1 | 1 |
| | | QIESLSLNGVPNIFLSTK | 2 | 1 | 1 | 1 | 1 | 1 |
| | | QIESLSLNGVPNIFLS | 2 | | | | 1 | |
| | | QIESLSLNGVPNIFL | 2 | | | | 1 | |
| | | IESLSLNGVPNIFLSTKASN | 2 | | | | | 1 |
| | | IESLSLNGVPNIFLSTKA | 2 | 1 | 1 | 1 | 1 | 1 |
| | | IESLSLNGVPNIFLSTK | 2 | 1 | 1 | 1 | 1 | 1 |
| | | IESLSLNGVPNIFLST | 2 | 1 | 1 | | 1 | 1 |
| | | IESLSLNGVPNIFLS | 2 | | | | 1 | |
| | | IESLSLNGVPNIFL | 2 | | | | 1 | |
| | | ESLSLNGVPNIFLSTKA | 2 | | | | | 1 |
| | | ESLSLNGVPNIFLSTK | 2 | | | | | 1 |
| | | SLSLNGVPNIFLSTKA | 2 | | | | 1 | |
| | | LSLKNFKALVYHVG | both | | | | | 1 |
| | | SLKNFKALVYHVGDTI | both | | | | | 1 |

**Fig 2. Peptides from class II molecules of bursal B cells infected in vitro.** Proteins and peptides in order of the genomic sequence; blue, green indicate multiple species from same peptide epitope; soluble BL linked with BLB2: *expressed, **not expressed; peptide species found in particular experiment indicated by 1.

nonamer motif that was tested did not produce a soluble complex with BL2*02 and therefore is likely to have been bound to a BL1*02 molecule on chicken bursal B cells.

## The overall structure of a chicken class II molecule is similar to those of mammals

Of the 4 soluble BL2*02 molecules linked to MDV peptides that were expressed, the one with a peptide from gE formed crystals that diffracted to sufficiently high resolution to determine a structure (Fig 3A–3C , S2 Table). As this work was being written up for publication, a paper was published with the crystal structure of a ribosomal peptide in complex with a soluble BL2 molecule from B19 [44], an MHC haplotype known to confer susceptibility to MD [13,45].

Overall, the structure of BL2*02 is extremely similar to BL2*19 and to the iconic mammalian class II molecule HLA-DR1 (Fig 3D and 3E), with root mean square deviations for Cα atoms of 0.684 and 0.878 Å, respectively (S3 Table). Most of the key residues found in mammalian class II molecules are identical in these 3 structures and found in nearly identical conformations, including those involved in domain folding (disulfide bonds and glycosylation site) and in interdomain, DM, and CD4 interactions (Fig 3E, S4 Fig, and S4 Table). However, both chicken molecules have a four amino acid extension to the first loop and second β-strand of the BLA α chain compared to HLA-DR1 (Fig 3D; [33,44]).

## BL2 from B2 illustrates a new mode of peptide binding for MHC class II molecules

The way in which peptides bind to mammalian class II molecules has been studied in detail, showing that a nonamer core binds as a polyproline II helix, with main chain atoms forming hydrogen bonds (H-bonds) with the MHC molecule and with 3 or 4 side chains binding into pockets that determine the specificity of different MHC alleles [27,28]. Typically, the bound peptides extend outwards from the N- and C-terminals of the core, permitted by an arginine found in all class II molecules and in the classical class I molecules of nonmammalian vertebrates and with these flanking regions sometimes assuming stable structures that can be recognized by T cells [27,46–48]. In fact, the P10 side chain of a peptide with a nonamer core has been reported [49] in one structure to bind onto a polymorphic shelf outside of the groove (S5 Fig).

In stark contrast to all reported mammalian class II structures and to BL2*19, the BL2*02 has a decamer core, with the structure showing side chains at peptide positions P1, P4, P5, P8, and P10 of the core binding into pockets, so that the peptides that were first analyzed with a nonamer motif were reanalyzed for this peptide motif (Figs 3A and 4A and 4B); the decamer sequence predicted by Gibbs clustering corresponds to the core sequence found in the structure. Compared to mammals and to BL2*19, the peptide in BL2*02 shifts the residues pointing up from the core towards the T cell from the typical P2, P3, P4, or P5 and P8 to P2, P3, P6, and P9, shifting register at P4 (Fig 4B). The class II residues that form H-bonds with the main chain atoms of the peptide were originally described for HLA-DR1 [27] and are identical in chickens (except for chicken Nα80 and Qβ57 which are R and D in humans, and Iβ71 in BL2*02 which is R in the other 2 structures), but again BL2*02 shifts register at P4, with subsequent interactions occurring 1 peptide position after HLA-DR1 and BL2*19 (Fig 4D). However, current analyses of such bonding are more refined, with some residues interacting through water molecules and other polymorphic residues contributing to main chain peptide interactions (S6–S8 Figs).

The conformation of the peptide bound to BL2*02 shows a polyproline II helix as in mammals but deviates downwards just at P4 and then returns to the helical conformation (Fig 4C). Comparing the peptides from the side view confirms that it is peptide position P4 that is located lower in the groove than in other reported class II peptides (Figs 3D and 4B, 4E, and

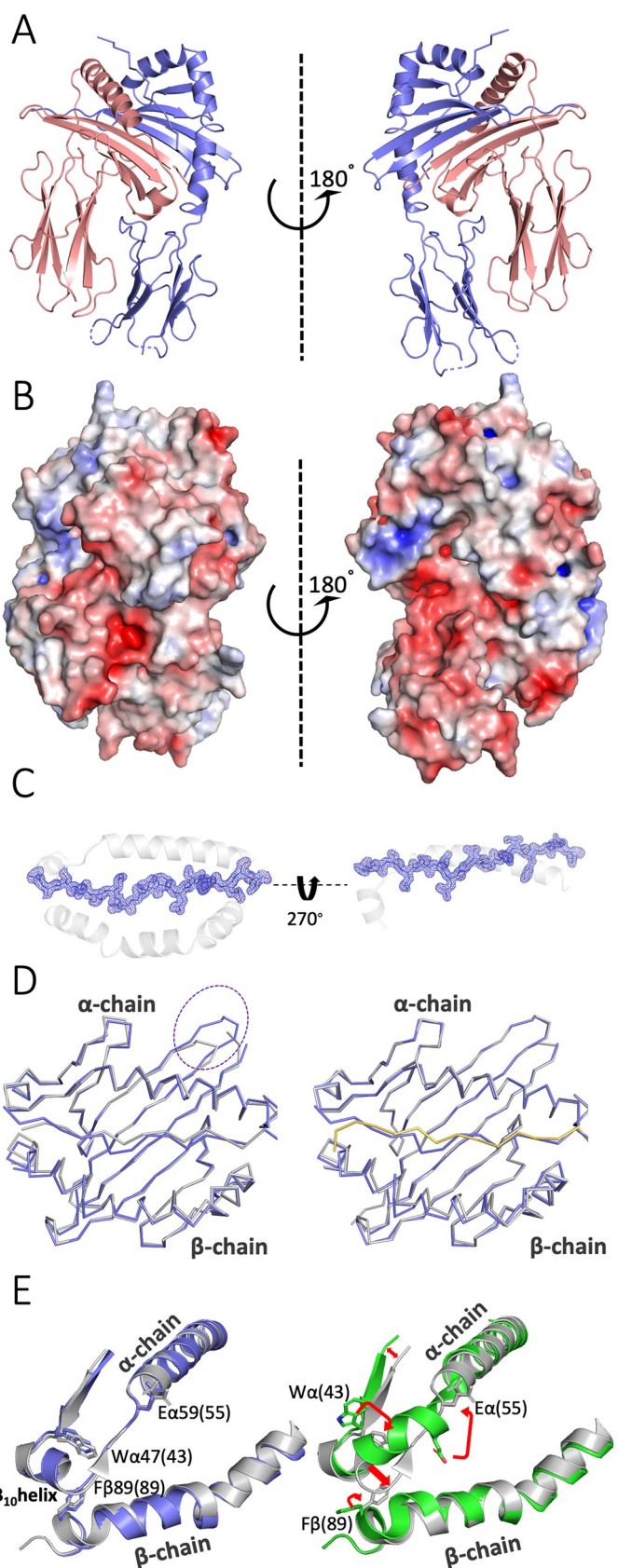

**Fig 3. The structure of the chicken BL2*02 molecule with peptide QIESLSLNGVPNIFLSTKA from gE is very similar to the BL2*19 molecule and the human HLA-DR1*01 molecule.** (A) Two side views of BL2*02 with peptide in ribbons; α chain and peptide, slate blue; β chain, deep salmon red. (B) Two side views of BL2*02 with peptide as solvent-accessible surface calculated by APBS electrostatics (positive charge, blue; negative charge, red). (C) Top view and side view (through β1 domain omitted for clarity) of BL2*02 in ribbons and peptide in stick as a composite omit $(2F_o\text{-}F_c)$ map contoured at 1 Å in blue chicken wire. (D) Top view of Cα backbone of class II molecule with peptides superimposed, α1 domain above and β1 domain below. Left panel: BL2*02, slate blue; DR1*01, grey; dotted circle indicates loop with four amino acid insert in chicken class II molecules. Right panel: BL2*02, slate blue; BL2*19, grey with peptide in yellow. (E) Top view of a portion of the class II molecules superimposed in ribbons, α1 domain above and β1 domain below, with key residues [Wα47(43), Eα59(55), Fβ89(89)] in sticks to show differences (including $3_{10}$ to α-helix) upon DM binding. Left panel: BL2*02, slate blue; DR1*01, grey. Right panel: DR1*01 (PDB accession 4X5W), white; DR1*01 in DM co-crystal (4GBX), green; red arrows indicate major structural movements. The underlying data for this figure can be found in PDB file 6T3Y.

4F). This "crinkle" in the peptide is due to a single BLB residue Serβ78 in BL2*02 that leads to a deeper pocket compared to Pheβ78 from BL2*19 and Tyrβ78 from HLA-DRB1, and which pulls P4 downwards and pulls the main chain register towards the N-terminus of the peptide, resulting in the decamer core (Fig 4B and 4E). Comparison of the contacts around P4 shows that this relatively small residue Serβ78 accommodates the peptide in the conformation adopted in BL2*02, while larger residues Tyrβ78 in HLA-DR1 and Pheβ78 in BL2*19 would conflict (Fig 4F).

## Discussion

The chicken MHC can determine life or death after infection from certain economically important pathogens, of which MDV is an iconic example. Infection of bursal B cells in culture allowed us to determine the peptides bound to class II molecules, the peptide motifs of the 2 class II molecules encoded by the B2 haplotype, the MDV genes that give rise to the pathogen peptides that are potential T cell epitopes, and an unexpected mode of peptide binding to the dominantly expressed class II molecule of this haplotype.

The size distribution, raggedy ends, enrichment of proline in the N-terminal flanking region, and multiple amino acids for anchor positions of the peptides bound to the class II molecules of the chicken MHC B2 haplotype are similar to mammalian class II molecules, but there were surprises with the motifs. The BL1*02 molecule (composed of BLA α chain and BLB1*02 β chain) has a motif with large aliphatic residues at P1 and aromatic and large aliphatic residues at P9, but only a slight enrichment for basic amino acids in between and without any obvious anchor residue preferences. The BL2*02 molecule (BLA α chain and BLB2*02 β chain), which is the dominantly expressed class II molecule in most tissues [26], has an unprecedented decamer core motif, with large aliphatic residues at P1 and P4, small and even acidic residues at P5, and large hydrophobic residues at P10.

It was also a surprise to find that, in this in vitro infection of chicken bursal B cells, only 4 MDV genes gave rise to 77% (49/64) of the class II-bound peptide species from the pathogen and 39% (9/23) of all the potential T cell epitopes identified. In fact, epitopes from these 4 MDV genes were found in many of the samples, whereas the most peptides were detected in only 1 sample and are likely to be very rare, so the impact of the epitopes from the 4 MDV genes will be much larger than the numbers suggest. The 4 genes all encode structural proteins present in the virion, 3 surface glycoproteins, and 1 polytopic tegument protein (S1 Table), but these genes are also expressed early after infection of B cells [50]. There are many other proteins present in the virion, and many other proteins expressed early after infection [14,16], so why these 4 proteins are the source of most presented peptides is not at all clear yet.

This dominance of just a few class II epitopes from a few MDV proteins stands in contrast to other human and mouse herpesviruses, for which polyclonal (and in some cases

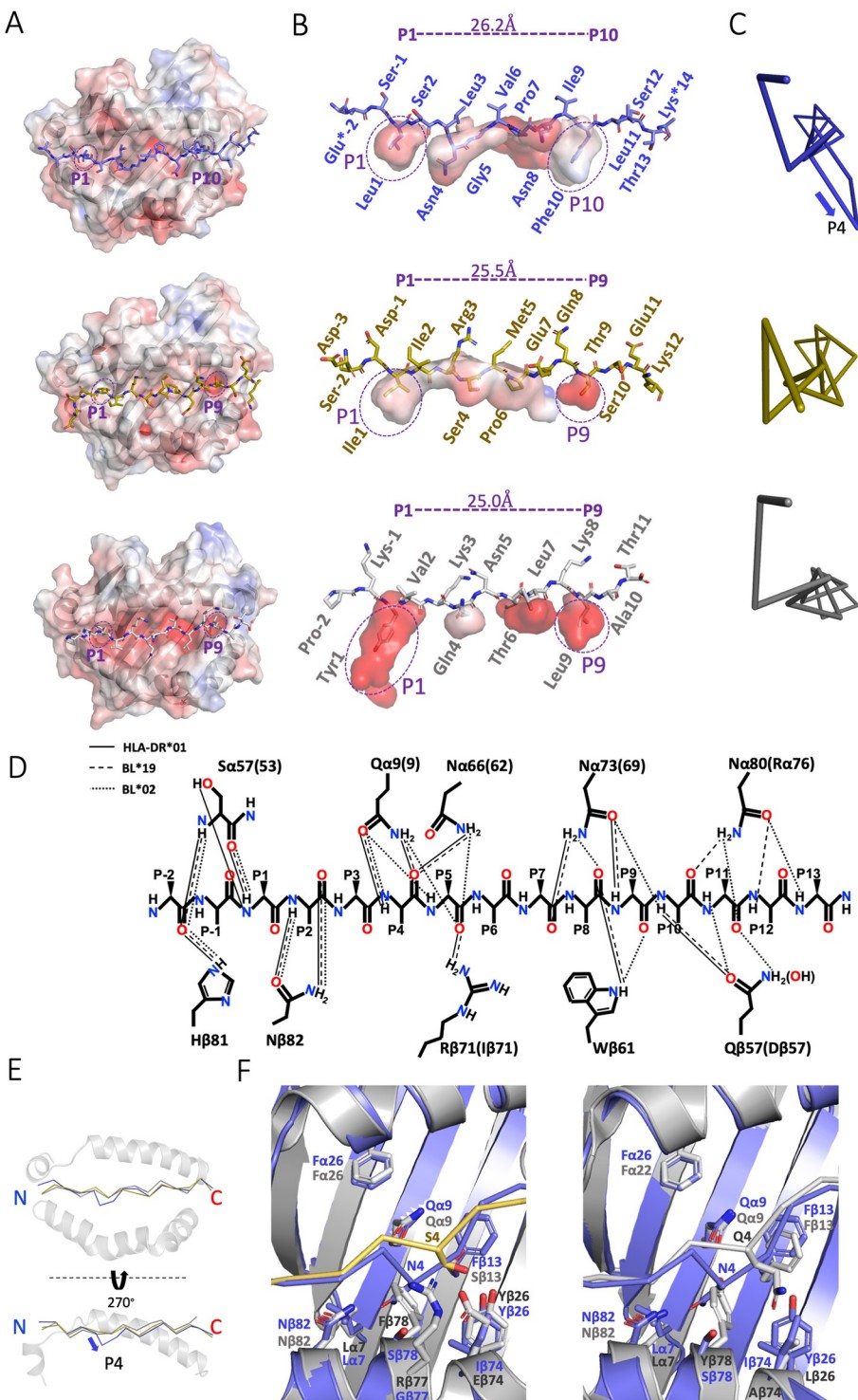

**Fig 4. Although similar in overall structure to other class II molecules, BL2*02 binds peptides with a decamer motif due to Serβ78.** (A–C) Top panels, BL2*02 (PDB accession 6T3Y); middle panels, BL2*19 (6KVM); bottom panels, DR1*01 (1DLH). (A) Top view of solvent-accessible surface of class II molecules calculated by APBS electrostatics (positive charge, blue; negative charge, red) with P1, P9, and P10 pockets indicated and with peptides in sticks. (B) Side view of peptide in sticks with amino acids indicated and with pockets as surfaces with P1, P9, and P10 pockets indicated, and with length between Cα of P1 and P9 or P10 shown (blue asterisks indicate that the side chains are not resolved). (C) Edge-on view of peptide Cα backbone, with arrow indicating departure of P4 in BL2*02 from the polyproline II helix. (D) Schematic of H-bonds interacting with peptide main chain atoms (cut-off of 3.2 Å) originally

described for DR1\*01 (solid lines) compared to BL2\*19 (dashed lines) and BL2\*02 (dotted lines); additional interactions subsequently found are shown in S6–S8 Figs). (E) Ribbon representation of class II molecule with superimposing peptide Cα backbones for BL2\*02 (slate blue), BL2\*19 (yellow), and DR1\*01 (grey) showing top view (α1 domain above, β1 domain below) and side view (α1 domain shown, β1 domain removed for clarity), and with arrow indicating departure of P4 in BL2\*02 from the polyproline II helix. (F) Close-up view of class II molecules in ribbons with peptide Cα backbones, and with the side chains for residues around pocket 4 and peptide P4 as sticks, superimposing BL2\*02 (slate blue) with BL2\*19 (grey with yellow peptide) in left hand panel and with DR1\*01 (grey) in right hand panel. The underlying data for this figure can be found in PDB file 6T3Y.

polyfunctional) CD4$^+$ T cell responses to many codominant epitopes have been reported, with multiple epitopes within some proteins [51–53]. The situation may also be true for vaccina virus [54,55], another virus with a similarly large DNA genome. However, this finding of few class II epitopes for MDV could fit with the existence of chicken MHC haplotypes that confer susceptibility, which would be difficult to explain with a larger number of T cell epitopes.

The overall structure of the chicken class II molecule is similar to that of mammals, which is perhaps not such a surprise given the high level of sequence identity, particularly in the BLA α chain [33]. Almost all the key features and residues involved in interactions with DM, CD4, and peptides are conserved. However, the details of the peptide binding to BL2\*02 were unexpected, all deriving from a single amino acid polymorphism in the BLB2\*02 β chain. The small MHC residue Serβ78 leads to a pocket which pulls in the side chain from peptide position P4, resulting in a crinkle in the peptide unlike the completely flat peptide in a polyproline II helix as is normally found. This crinkle pulls the C-terminus of the peptide towards the N-terminus, resulting in a shift of 1 peptide position along for the main chain interactions, the anchor residues (P1, P4, P5, P8, and P10) and the side chains pointing up towards the T cell (P2, P3, P6, and P9). It has long been known that differences in the size of a single class II side chain can have strong effects on which peptides are bound and thus on the subsequent functional effects, as in the example of the Val-Ile dimorphism of β89 in HLA-DRB1 [56].

The MHC haplotype B2 is known to confer resistance to MD (as well as other common poultry pathogens [7]). As mentioned above, a structure was very recently reported for a chicken class II molecule from B19 [44], a haplotype that is known to confer susceptibility to MD [13,45]. This B19 molecule has a nonamer peptide motif, which opens the possibility that the length of core peptide might contribute to differences in resistance to MD. However, the chicken MHC (the BF-BL region) has low rates of recombination [57–59], meaning that a number of genes are inherited as haplotypes. Thus, much more work is required to determine whether class II molecules are important as suggested [23–25], along with the classical class I genes (BF1 and BF2), the NK receptor/ligand pair (BNK and Blec), and the butyrophilin homolog (BG1) which all have been proposed as candidates to explain the strong genetic associations with infectious pathogens [19–22].

No matter what the contribution of chicken class II molecules to differential resistance to MDV infection, the design and results of this study open the possibility of understanding T cell epitopes to generate improved vaccines for MDV as well as other economically important and zoonotic poultry pathogens [6,8–11,17,18]. Moreover, the results from this relatively simple system for chicken MHC haplotypes have relevance for human viral disease, given the class I molecules encoded by individual loci in mammals may function like the dominantly expressed class I molecule of chickens [7].

## Methods

### MDV infection of bursal B cells in culture

**Viruses.**  MDV reporter viruses expressing the green fluorescent protein (GFP) under the control of the herpes simplex virus 1 (HSV-1) thymidine kinase promotor were generated

based on the very virulent RB-1B field strain (GenBank accession number EF523390) and the vaccine strain CVI988 (DQ530348). GFP was inserted into the bacterial artificial chromosome (BAC) backbone replacing the Eco-gpt gene [60]. BACs were confirmed by Illumina MiSeq sequencing to verify sequence integrity.

Chicken embryo cells (CEC) were isolated from 11-day-old VALO specific-pathogen-free (SPF) embryos (VALO Biomedia; Osterholz-Scharmbeck, Germany) as described previously [61]. CEC were maintained in minimal essential medium (MEM, PAN Biotech; Aidenbach, Germany) supplemented with 1% to 10% fetal bovine serum, 100 U/ml penicillin, and 100 μg/ml streptomycin (AppliChem; Darmstadt, Germany) at 37°C and 5% $CO_2$.

All recombinant GFP reporter viruses were reconstituted by transfection of fresh CEC with purified BAC DNA using $CaPO_4$ transfection [62]. The viruses were propagated on CEC for up to 9 passages, and infected cells were stored in liquid nitrogen.

**Isolation and culture of bursal B cells.** Fertilized eggs of M11 (B2/B2) chickens were kindly provided by Dr. S. Weigend (Federal Research Institute for Animal Health, Mariensee, Germany) and hatched at the Faculty for Veterinary Medicine, Munich. Birds were housed under conventional conditions in aviaries with groups up to 10 birds and received food and water ad libitum. Bursas were isolated from 6- to 8-week-old birds, and cells were obtained by dissociation of the organs in PBS using a stainless steel sieve. Leukocytes were isolated by density gradient centrifugation on Biocoll (1.077 g/ml, Biochrom, Berlin, Germany).

B cells were cultured at 40°C in Iscove's modified Dulbecco's medium (IMDM), 100 U/ml penicillin, 100 μg/ml streptomycin, 8% (vol/vol) fetal bovine serum (all Bio&Sell, Nürnberg, Germany), and 2% (vol/vol) chicken serum (ThermoFisher Scientific, Waltham, USA), with the addition of recombinant soluble chicken BAFF and chicken CD40L, as described [37,63,64].

**Infection of bursal B cells.** For MDV infection, $1 \times 10^7$ B cells were co-cultured with $3 \times 10^5$ freshly thawed MDV-infected CECs (representing an infectious dose between 0.5 to $1 \times 10^5$ plaque-forming units) in 1 ml in a 24-well plate. Up to $10^9$ cells were harvested 24 hours after infection, washed with PBS 3 times, and frozen at −80°C. At this time point, cultures contained 25% to 35% viable B cells and 10% to 15% of viable B cells were GFP positive (infected with MDV), as assessed by flow cytometry with the Fixable Viability Dye eFluor 780 (Thermo Fisher Scientific) using a FACSCanto II flow cytometer and FACS DIVA and FlowJo software (BD, Heidelberg, Germany). Parallel flow cytometry experiments showed that the CECs did not express class II molecules (S9 Fig).

**Ethical permission.** Permission for the procedures involving animals was given by Landesamt für Gesundheit und Soziales (LAGeSo) in Berlin (approval number T0245/14) and in Munich (file number KVR-I/221-TA 160/13-He, Inst.-Nr. 01-17a) with organ sampling in accordance with the German Animal Welfare Act.

## Isolation of class II molecules and immunopeptidomics

**Isolation of peptides from class II molecules.** As described [38,65], frozen cells were thawed into 10 mM CHAPS (AppliChem, St. Louis, Missouri) in PBS (Gibco, Carlsbad, California) with complete protease inhibitor (Roche, Basel, Switzerland), and class II molecules were isolated by standard immunoaffinity purification using purified monoclonal antibody 2G11 [66] produced in-house and covalently linked to CNBr-activated Sepharose (GE Healthcare, Chalfont St Giles, UK). MHC–peptide complexes were eluted by repeated addition of 0.2% trifluoroacetic acid (TFA; Merck, Whitehouse Station, New Jersey). Peptides were purified by ultrafiltration using centrifugal filter units (Amicon; Millipore, Billerica, Massachusetts), desalted using ZipTip C18 pipette tips (Millipore), and eluted in 35 μl 80% acetonitrile

(Merck), 0.2% TFA, then vacuum-centrifuged and resuspended in 25 μl 1% acetonitrile, 0.05% TFA, and stored at −20°C.

**Analysis of peptides by LC–MS/MS.** Peptides were separated by reverse-phase liquid chromatography and analyzed from 2 to 3 technical replicates, with sample shares of 33% or 50% trapped on a 75 μm × 2 cm trapping column (Acclaim PepMap RSLC; Thermo Fisher) at 4 μl/min for 5.75 minutes. Peptide separation was performed at 50°C and a flow rate of 175 nl/min on a 50 μm × 25 cm separation column (nano-UHPLC, UltiMate 3000 RSLCnano; Thermo Fisher, Waltham, Massachusetts) applying a gradient ranging from 2.4% to 32.0% of acetonitrile over the course of 90 minutes. Samples were analyzed as previously described [67] using an online-coupled Orbitrap Fusion Lumos mass spectrometer (Thermo Fisher, Waltham, Massachusetts), implementing a top-speed CID method with survey scans at 120k resolution and fragment detection in the Orbitrap (OTMS2) at 60k resolution. The mass range was limited to 300 to 1,500 m/z with precursors of charge states greater than or equal to 2 eligible for fragmentation.

**Database search and spectral annotation.** LC–MS/MS results were processed using Proteome Discoverer (v.1.3 and 1.4; Thermo Fisher) to perform database search using the Sequest search engine (Thermo Fisher) with the chicken and appropriate MDV proteomes as reference database annotated by the UniProtKB/Swiss-Prot (www.uniprot.org), status April 2018. The search combined data of 3 technical replicates, was not restricted by enzymatic specificity, and oxidation of methionine residues was allowed as dynamic modification. Precursor mass tolerance was set to 5 ppm, and fragment mass tolerance to 0.02 Da. False discovery rate (FDR) was estimated using the Percolator node [68] and was limited to 5%. Length of peptides was limited to 12 to 25 amino acids. The mass spectrometry proteomics data have been deposited to the ProteomeXchange Consortium via the PRIDE [69] partner repository with the dataset identifier PXD023954. Possible motifs for peptides of length 10 amino acids or greater were grouped using GibbsCluster-2.0 [70], specifying parameters for class II ligands, number of groups, length of motif, and first residue as a hydrophobic amino acid; the process collapses all species with posttranslational modifications like oxidized methionines to one entry but keeps any overlapping or nested sequences as separate entries. Venn diagrams were done by BioVenn [71].

## Expression of class II molecules and structural determination

**Cell lines.** Sf9 insect suspension cells (*Spodoptera frugiperda* female ovarian cell line, ATCC CRL-1711) were used for production of baculovirus, and High Five insect suspension cells (*Trichoplusia ni* female ovarian cell line, GIBCO B85502) streptomycin were used for protein production, both grown in Insect-XPRESS Protein-free Insect Cell Medium (Lonza, BE12-730Q) supplemented with L-glutamine (to 1%), 50 U/ml penicillin, and 50μg/ml streptomycin at 27°C with 135 rpm shaking.

**Cloning, protein expression, and purification.** For protein production and crystallography, the different constructs were cloned separately into the baculovirus insect cell expression vector, pFastBac1 (GIBCO 10360014), modified to contain an N-terminal GP67 secretion signal sequence (for amino acid sequences of each portion, see S10 Fig). For the BLA construct, the GP67 signal sequence was followed by the ectodomain of the α-chain of BLA (Hα5 to Eα188; GenBank accession number AY357253, but with S79 due to dimorphism noted in [33]), then a short Gly/Ser-linker (G4S), a c-fos dimerization domain [72], a two amino acid linker (GT), and an Avi-tag sequence for biotinylation. For the BLB2*02 constructs, the GP67 signal sequence was followed by a peptide sequence (described in S10 Fig), a 15 amino acid Gly/Ser-linker [3(G4S)], the ectodomain of the β-chain of BLB2*02 (Sβ4 to Kβ198; AB426141),

a TEV-protease cleavage site, a c-jun dimerization domain [72], a Gly/Ser-linker [2(G4S)], a V5-tag, and a 6x His-tag. Correct cloning was confirmed by DNA sequencing.

To generate bacmids, the constructs were transformed DH10Bac cells (GIBCO 10361012), plated out and clones grown up overnight. Bacmid DNA was subjected to PCR using M13 primers to confirm transposition of inserts, and then transfected into Sf9 cells as follows to generate P0 viral stocks. First, 20 μl bacmid DNA and 15 μl Fugene (Promega E2311) were each diluted into 600 μl of Lonza Insect Xpress medium, mixed together, and incubated at room temperature for 20 minutes, and then 200 μl of the mixture were transferred to a well of a 6-well plate containing $9 \times 10^5$ Sf9 cells in 2 ml antibiotics-free medium and incubated for 4 to 6 days at 27°C. The cell medium containing baculovirus was collected, passed through a 0.2-μm filter (Sartorius 16532-K), and then this P0 stock was used to amplify the number of viruses to give the P1 stock as follows. First, 1 ml P0 stock was added to 50 ml of $2 \times 10^6$ Sf9 cells/ml in suspension and cultured at 27°C for 48 hours with shaking at 135 rpm, then the cells were pelleted, and the culture media was filtered as above. Both P0 and P1 viral stocks were stored at 4°C until needed.

High Five cells were grown in 0.6 L medium in 2 L flasks and transduced with 6 to 10 ml P1 stock. Forty-eight to 72 hours after infection, culture media was collected and incubated overnight at 4°C with nickel-Sepharose Excel (GE Healthcare 17-3712-01). His-tagged proteins were eluted from the nickel-Sepharose with 1 M imidazole in TrisCl (pH 8.5), then purified by FPLC size-exclusion chromatography using Superdex S200 in 100 mM TrisCl (pH 8.5), and subjected to endoproteinase Glu-C (V8 Protease, Sigma, United Kingdom) cleavage at 37°C overnight to remove C-terminal tags and dimerization domains, with successful proteolysis confirmed by size shift in SDS-PAGE. The proteins were again purified as above using Superdex S200 in 25 mM TrisCl (pH 8.5), and then concentrated using Amicon Ultra (with 3,000 molecular weight cut-off, Merck UFC8003) to 3 to 12 mg/ml as determined using a nanodrop spectrophotometer.

**Crystallization and structure determination.** Crystallization conditions were screened using the PEGs II suite (Qiagen, United Kingdom) at 20°C, with10.45 mg/mL protein in 25 mM TrisCl (pH 8.5) mixed at 1:1 or 1:2 with mother liquor to give 0.6 to 0.9 μl sitting drops. The protein complex BL2*02 with the peptide QIESLSLNGVPNIFLSTKA from MDV protein gE crystalized in 100 mM TrisCl (pH 8.5), 200 mM sodium acetate, and 30% w/v PEG 4000. Crystals were cryo-cooled in mother liquor supplemented with 10% to 35% (v/v) glycerol before data collection.

Diffraction data were collected remotely on the I04-1 beamline (Diamond Light Source, Oxford, UK) at a wavelength of 0.978 Å. Data reduction and scaling were performed using XDS [73] and SCALA [74]. The crystal of the BL2*02 belongs to the C 1 2 1 space group, and the structure was solved by basic molecular replacement deploying Phaser from the CCP4i package [75] using HLA-DR1 (4X5W) as the search model [76]. Further rounds of manual model building and refinement were done using Coot [77] and Phenix [78]. Further details about collection and refinement are shown in S2 Table. The structure was deposited in the Protein Database (PDB) on 14 October 2019 and assigned accession number 6T3Y.

**Interaction analysis.** LigPlot+ [79] was used to calculate the potential interactions from the crystal structure. Unless stated otherwise, the maximum hydrogen-acceptor distance was set to 2.70 Å and the maximum donor-acceptor distance was set to 3.35 Å. The minimum and the maximum contact distances of hydrophobic residue to any contact were set to 2.90 Å and 3.90 Å, respectively. PyMOL (The PyMOL Molecular Graphics System, Version 2.0 Schrödinger, LLC) was used to display and analyze the structural data visually.

**Differential scanning fluorimetry.** Differential scanning fluorimetry was performed using a CFX Connect real-time PCR thermal cycler (Biorad, United Kingdom) set to

Fluorescence Resonance Energy Transfer (FRET) scan mode. SYPRO orange (Invitrogen S6651) was diluted to 1:500 in 25 mM TrisCl (pH 8) and 12.5 μL mixed with 12.5 μL 10 μM protein in 25 mM TrisCl (pH 8) in each well of a 96-well plate, with buffer alone serving as negative control. The samples were scanned every 0.5°C from 25 to 95°C. Data analysis was performed in Prism (GraphPad San Diego, California, United States of America).

## Supporting information

**S1 Fig. There is a large overlap in peptides found in different samples, presented as Venn diagrams.** Comparison of peptides (A) from all CVI988 samples pooled and from all RB-1B samples pooled, (B) from CVI988 sample and RB-1B sample from 2019 experiment, (C) from CVI988 sample and RB-1B sample from 2020 experiment, (D) from CVI988 samples in 2019 and 2020 experiments, and (E) from RB-1B samples in 2018, 2019, and 2020 experiments. Total peptide numbers are below each sample name; percentages indicate peptides unique to a particular sample (that is, not shared) in a particular comparison and are rounded. The underlying data for this figure can be found in S1–S6 Data.
(PDF)

**S2 Fig. Length distribution of peptides found in the 6 samples.** Top panel, absolute numbers of peptides for each length; bottom panel, percentage of total peptides for each length. The underlying data for this figure can be found in S1–S6 Data.
(PDF)

**S3 Fig. The N-termini of the class II β chains of humans and chickens are in the same position.** Shown are portions of the genomic sequences of HLA-DRB1 (AM910430 [80]) and BLB2 (called BLBII in M29763 [43]), similar to sequences from other class II molecules ([34] and other papers cited therein), aligned with the N-terminal sequences of DR1 β chain [81], similar to other human class II β chains (reviewed in [82]). Nucleotide sequences in lower case along with numbering from GenBank flat files (AM910430 for HLA-DRB1, M29763 for BLB2) with introns in lower case italic, invariant residues at start and end of introns in bold and underlined. Protein sequences in capital letters either from translation of genomic sequence or N-terminal amino acid sequencing, with signal sequence cleavage site double underlined (60% probability from SignalP-5.0, www.cbs.dtu.dk/services/SignalP/) and possible downstream signal sequence cleavage site single underlined.
(PDF)

**S4 Fig. Key residues in chicken and human class II molecules are identical or similar.** Upper panel, class II α chains; lower panel, class II β chains. Single letter amino acid code; blue, identical residues; green box, CD4 contact; yellow box, DM contact; grey box, hydrophobic transmembrane region; blue box, cytoplasmic tail. Arrow, β-strand; blue cylinder, α-helix; purple cylinder, 3₁₀-helix (secondary structure elements as determined by PyMol); red "SS," cysteine for intrachain disulfide bond; orange "g," glycosylation site. The underlying data for this figure can be found in PDB files 1DLH, 6KVM, and 6T3Y.
(PDF)

**S5 Fig. The only example reported for peptide position P10 binding a class II molecule is not a deep pocket but a shelf, which is outside of the groove that binds the nonamer core (49).** Comparison of BL2*02 (PDB file 6T3Y) in upper panel with DR1 (1T5W) in lower panel for (A) top view of solvent-accessible surface of class II molecules calculated by APBS electrostatics (positive charge, blue; negative charge, red) with P1 and P10 pockets for BL2*02 indicated (upper panel) and the P1 and P9 pockets along with the P10 shelf for DR1 indicated

(lower panel), and with peptides in sticks, (B) side view of peptide in sticks with amino acids indicated and with pockets as surfaces with P1 and P10 pockets indicated for BL2*02, and with length between Cα of P1 and P9 or P10 shown (blue asterisks indicate that the side chains are not resolved), and (C) edge-on view of peptide Cα backbone, with arrow indicating departure of P4 in BL2*02 from the polyproline II helix. (D) Ribbon representation of class II molecule with superimposing peptide Cα backbones for BL2*02 (slate blue) and DR1 (1T5W, grey) showing top view (α1 domain above, β1 domain below) and side view (α1 domain shown, β1 domain removed for clarity), and with blue arrow indicating departure of P4 in BL2*02 from the polyproline II helix. The underlying data for this figure can be found in PDB files 1T5W and 6T3Y.
(PDF)

**S6 Fig.** H-bonds (dotted lines, cut-off of 4 Å) between class II molecules (ribbons with side chains of key interacting residues in brown sticks, with residue numbers in green), peptide (purple) and water molecules (blue circles), as well as hydrophobic contacts (red spokes with residue numbers in black), as determined by LigPlot+ v.2.2 (www.ebi.ac.uk/thornton-srv/software/LigPlus/), based on structures 1DLH for DR1*01, 6KVM for BL*19, and (A) 6T3Y for BL*2; (B) 6KVM for BL*19; (C) 1DLH for DR1*01; and (D) 4X5W also for DR1*01. The underlying data for this figure can be found in PDB files 1DLH, 4X5W, 6KVM, and 6T3Y.
(PDF)

**S7 Fig. 3D representation of H-bonds (dotted lines, cut-off of 4 Å) between class II molecules (ribbons with side chains of key interacting residues in sticks), peptide in sticks, and waters as blue circles.** Top panel, BL2*02 (6T3Y) with peptide in blue; middle panel, BL2*19 (6KVM) with peptide in yellow; bottom panel, HLA-DR1*01 (1DLH) with peptide in grey. Based on LigPlot analyses in S6 Fig. The underlying data for this figure can be found in PDB files 1DLH, 4X5W, 6KVM, and 6T3Y.
(PDF)

**S8 Fig.** Schematic of H-bonds interacting with peptide main chain atoms (cut-off of 4 Å) comparing HLA-DR1*01 (solid lines), BL2*19 (dashed lines), and BL2*02 (dotted lines), including H-bonds through waters (solid blue circles), based on structures 1DLH for DR1*01, 6KVM for BL*19, and 6T3Y for BL*2 analyzed by LigPlot (S6 Fig). The underlying data for this figure can be found in PDB files 1DLH, 4X5W, 6KVM, and 6T3Y.
(PDF)

**S9 Fig. Flow cytometric analysis CECs used for infection of bursal B cells shows virtually no expression of chicken class II molecules.** Dot plots (GFP from MDV on x-axis, PE from antibody staining on the y-axis) for (A) chicken class II molecules using the monoclonal antibody 2G11 and (B) isotype control antibody. (C) Histogram of antibody staining (mean fluorescent intensity on x-axis, number of events on the y-axis); red, 2G11 staining; blue, isotype control. CECs, chicken embryo cells; GFP, green fluorescent protein; MDV, Marek's disease virus; PE, phycoerythrin; TEV, tobacco etch virus.
(PDF)

**S10 Fig. Protein sequences encoded by constructs for soluble class II chains in Baculoviruses.** Sequences in single letter amino acid code, all black except: bold blue, TEV protease cleavage site; bold oranges, c-fos tag; bold gold, c-jun tag; bold red, V5-tag; normal red, Avi-tag (biotinylation site). Highlights: green, GP67 signal sequence; deep pink, peptides; cyan, linkers; grey, MHC extracellular domains; yellow, 6xHis-tag. MHC, major histocompatibility

complex; TEV, tobacco etch virus.
(PDF)

**S1 Table. Peptides from class II molecules of bursal B cells infected in vitro.** The underlying data for this figure can be found in Fig 2 and S1–S6 Data.
(PDF)

**S2 Table. Crystallographic statistics.** The underlying data for this figure can be found in PDB file 6T3Y.
(PDF)

**S3 Table. Root mean square deviation (RSMD) of BL2*02 (6T3Y) against chains and domains from 3 other class II structures.** The underlying data for this figure can be found in PDB files 1DLH, 4X5W, 6KVM, and 6T3Y.
(PDF)

**S4 Table. Amino acid residues of BL*02 (6T3Y), BL*19 (6KVM), and HLA-DR1 (1DLH and 4X5W) that form H-bonds to the main chain of the peptide as determined by LigPlot +, with a maximum hydrogen-acceptor distance of 2.70 Å, a maximum donor-acceptor distance of 3.35 Å, and minimum and maximum contact distances for hydrophobic residue to any contact of 2.90 Å and 3.90 Å, respectively. Bridging water molecules are designated by "= HOH." The underlying data for this figure can be found in PDB files 1DLH, 4X5W, 6KVM, and 6T3Y.**
(PDF)

**S1 Data. Peptide ligands eluted from class II molecules isolated from bursal B cells from M11 chickens, infected with RB-1B in 2018.**
(XLSX)

**S2 Data. Peptide ligands eluted from class II molecules isolated from bursal B cells from M11 chickens, uninfected in 2019.**
(XLSX)

**S3 Data. Peptide ligands eluted from class II molecules isolated from bursal B cells from M11 chickens, infected with RB-1B in 2019.**
(XLSX)

**S4 Data. Peptide ligands eluted from class II molecules isolated from bursal B cells from M11 chickens, infected with CVI988 in 2019.**
(XLSX)

**S5 Data. Peptide ligands eluted from class II molecules isolated from bursal B cells from M11 chickens, infected with RB-1B in 2020.**
(XLSX)

**S6 Data. Peptide ligands eluted from class II molecules isolated from bursal B cells from M11 chickens, infected with CVI988 in 2020.**
(XLSX)

## Acknowledgments

We thank Dr. Alicia Martin Lopez for constructs that were the source of the BLA and BLB sequences, Dr. Paul Brear of the Crystallography facility in the Department of Biochemistry at the University of Cambridge for much help, the staff of beamline I04 at the Diamond Light

Source for help with data collection, and Dr. Clive Tregaskes, Dr. Hassnae Afrache, Prof. Gillian Griffiths, Prof. Lonneke Verwelde, and Prof. Tobias Lenz for critical reading of the manuscript.

## Author Contributions

**Conceptualization:** Bernd Kaspers, Sonja Härtle, Jim Kaufman.

**Data curation:** Samer Halabi, Michael Ghosh, Sonja Härtle, Jim Kaufman.

**Formal analysis:** Samer Halabi, Michael Ghosh, Sonja Härtle, Jim Kaufman.

**Funding acquisition:** Stefan Stevanović, Hans-Georg Rammensee, Benedikt B. Kaufer, Bernd Kaspers, Jim Kaufman.

**Investigation:** Samer Halabi, Michael Ghosh, Luca D. Bertzbach, Sonja Härtle, Jim Kaufman.

**Project administration:** Jim Kaufman.

**Resources:** Stefan Stevanović, Hans-Georg Rammensee, Luca D. Bertzbach, Benedikt B. Kaufer, Martin C. Moncrieffe, Bernd Kaspers, Jim Kaufman.

**Supervision:** Stefan Stevanović, Hans-Georg Rammensee, Benedikt B. Kaufer, Bernd Kaspers, Jim Kaufman.

**Validation:** Samer Halabi, Michael Ghosh, Martin C. Moncrieffe, Jim Kaufman.

**Visualization:** Samer Halabi, Michael Ghosh, Jim Kaufman.

**Writing – original draft:** Samer Halabi, Michael Ghosh, Luca D. Bertzbach, Benedikt B. Kaufer, Sonja Härtle, Jim Kaufman.

**Writing – review & editing:** Samer Halabi, Michael Ghosh, Stefan Stevanović, Hans-Georg Rammensee, Luca D. Bertzbach, Benedikt B. Kaufer, Martin C. Moncrieffe, Bernd Kaspers, Sonja Härtle, Jim Kaufman.

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
