## [Editor Report · Decision Letter 0]

19 Nov 2020

Dear Dr. Kaufman, 

Thank you for submitting your manuscript entitled "Differential host response to a herpesvirus: Marek’s disease virus peptides on chicken MHC class II molecules are derived from only a few genes and illustrate a new mode of peptide binding" for consideration as a Research Article by PLOS Biology.

Your manuscript has now been evaluated by the PLOS Biology editorial staff [as well as by an academic editor with relevant expertise] and I am writing to let you know that we would like to send your submission out for external peer review.

Please re-submit your manuscript within two working days, i.e. by Nov 21 2020 11:59PM.

Kind regards,

Paula

---

Associate Editor

PLOS Biology

---

## [Decision Letter · Decision Letter 1]

29 Jan 2021

Dear Dr. Kaufman,

Thank you very much for submitting your manuscript "Differential host response to a herpesvirus: Marek’s disease virus peptides on chicken MHC class II molecules are derived from only a few genes and illustrate a new mode of peptide binding" for consideration as a Research Article at PLOS Biology. Your manuscript has been evaluated by the PLOS Biology editors, an Academic Editor with relevant expertise, and by several independent reviewers.

In light of the reviews (below), we are pleased to offer you the opportunity to address the comments from the reviewers in a revised version that we anticipate should not take you very long. We will then assess your revised manuscript and your response to the reviewers' comments and we may consult the reviewers again.

In particular, reviewer #1 wants you to add further details in the parameters used for Gibbs clustering, asks whether you have controlled for MHCII expression by the CECs, and wants you to clarify text and figures. Reviewer #2 says wants you to shorten the manuscript, clarify the text and make it more accessible to non-specialists. Please address all the comments from reviewers. Please also make sure to address the data and other policy-related requests noted at the end of this email.

We think that this manuscript will fit as a PLOS Biology Discovery Report. Discovery Reports describe novel and intriguing initial findings with the potential to lead to a significant new result for the field. Discovery Reports are short articles, typically with 2-4 main figures. While the research may be preliminary, studies should be advanced to the stage where observations or findings have been confirmed by independent methods or experimental approaches and obvious alternative interpretations have been ruled out. Discovery Reports are designed to work together with Update Articles to empower researchers to evaluate and share work in a way that more closely mirrors the real-world research process and create a comprehensive research story. We think that the identification of a new mode of MCH class II binding and the immunodominance of only a few proteins seem well suited to a Discovery Report, which could eventually be updated with insight into epitope selection or the basis for the immunodominance. You can choose this Article type when re-submitting your work.

We also suggest a title change in order to make it simple and highlighting the message. Our suggestions are:

"Chicken MHCII uses a new binding mode to present Marek disease herpesvirus peptides" or "Strong immunodominance of epitopes derived from only four Marek disease herpesvirus proteins that are presented by chicken MHCII using a new binding mode". Feel free to modify our suggestions or use another title.

We expect to receive your revised manuscript within 1 month.

**IMPORTANT - SUBMITTING YOUR REVISION**

1. A cover letter that should detail your responses to any editorial requests.

2. A 'Response to Reviewers' file - this should detail your responses to the editorial requests, present a point-by-point response to all of the reviewers' comments, and indicate the changes made to the manuscript.

3. In addition to a clean copy of the manuscript, please also upload a 'track-changes' version of your manuscript that specifies the edits made. This should be uploaded as a "Related" file type.

*Resubmission Checklist*

*Published Peer Review*

*Early Version*

*PLOS Data Policy*

*Blot and Gel Data Policy*

Sincerely,

Paula

---

Associate Editor,

pjaureguionieva@plos.org,

PLOS Biology

DATA POLICY:

Regardless of the method selected, please ensure that you provide the individual numerical values that underlie the summary data displayed in the following figure panels as they are essential for readers to assess your analysis and to reproduce it: Figures 1A, 1B, 2, Supplementary Figure 1A, 1B, 1C, 1D, 1E and Supplementary Figure 2.

**Please also ensure that figure legends in your manuscript include information on where the underlying data can be found, including the structural data, and ensure your supplemental data file/s has a legend.**

**Please ensure that your Data Statement in the submission system accurately describes where your data can be found.**

REVIEWS:

Reviewer #1: Immunopetidomics.

Reviewer #2: Structural immunology.

Reviewer #1: Here Halabi et al present a study characterising peptide presentation by the chicken MHC class II of the B2 haplotype in a Marek's disease virus (MDV) infection model, incorporating both immunopeptidomics and structural biology. Whilst mammalian MHC class II have been broadly studied, data on chicken MHC II are comparatively sparse. Only a handful of studies have utilised mass spectrometry to characterise peptide presentation by chicken MHCII, revealing relatively few ligands. Furthermore, only one structure of a chicken MHCII (BL2*19) has been published prior to this study. Thus, Halabi et al. greatly advance knowledge of peptide presentation by chicken MHCII, identifying over 8000 naturally presented peptides through LC-MS/MS analysis of peptides eluted from immunoaffinity purified MHCII, and pairing these data with structural characterisation of BL2*02 to define the MHCII binding motif. Moreover, combining these techniques, they reveal novel MDV-derived ligands and a novel mode of MHCII binding. These data will be of great interest for vaccine development in the poultry industry, as well as to those in broader immunological research, from both an evolutionary and mechanistic perspective of antigen presentation.

Major comments:

1. Gibbs clustering is used to dissect the binding motifs of the co-expressed BL1 and BL2 and lines 157 and 158 state "the ratio of peptides from these ex vivo bursal B cells was around 40% nonamer and 60% decamer in all samples (Datasets S1-S6)." However, whilst table 1 notes which MDV peptides were assigned to either cluster 1 or cluster 2, this is not shown in the full Datasets S1-S6, making it hard to determine the ratio as stated. Nor are the numbers of peptides on which each motif is generated shown in Figure 1. To enable evaluation by the reader, further details on the parameters used for Gibbs clustering (including pre-processing such as whether overlapping peptide species were collapsed into single epitopes prior to clustering, if datasets were analysed by gibbs clustering individually or collectively) should be incorporated into the methods. Assignment to cluster 1 or 2 could be added to the Datasets file, and/or the number of peptides on which the motifs were generated added to figure 1.

2. For infection, a co-culture system is used in which there are 3 infected CECs to every 100 B2/B2 bursal cells. Has MHCII expression by the CECs been considered/controlled for? Could this be contributing to peptides isolated, thus impacting the motif characterisation? Is the MHC haplotype of the CECs known?

Minor comments

3. In the legend for figure 1, it would be useful to indicate the region of each peptide that is considered to be the decamer or nonamer motif

4. For clarity, would it be appropriate to label gH in Figure 2, for those not familiar with the virus like myself? Or include UL22 in the main text?

5. In table 1, are any of the detected peptides unique to a given strain? It is not currently commented upon whether all peptides are within the proteomes of both viral strains used.

6. In table 1 and the supplementary datasets, are lower case "m"s oxidised methionine? Could this be defined?

7. Line 261 - structure is missing a c

8. Line 323 - I only counted 9/23, rather than 10/23?

9. Line 432 - "Analysis of peptides by LC- and below MS/MS." Should "and below" be removed?

10. Line 442 - the mass range is stated as 400-650m/z, precursors 2+ and 3+. This seems a bit narrow. Should 650 read 1650?

Reviewer #2: Unlike their human counterparts, chicken MHC class II genes have strong associations with infectious disease and are thus of critical importance for the poultry industry. A major poultry pathogen is an oncogenic herpes virus that causes Marek's disease and has a genome comprising >100 genes. The B2 MHC haplotype correlatedswith susceptibility to Marek's. Using new technology, Halabi et al have have comprehensively characterized the peptides that are bound by the MHC class II isoforms encoded by the B2 haplotype. The vast majority of these peptides derive from the products of only four genes, and most are presented by BL2, the dominant MHC class II of the chicken. Whereas most MHC class II isoforms engage a nonamer sequence within an antigenic peptide, BL2 engages a decamer. Crystallographic analysis shows that BL2 engages a decamer. This is achieved by a small amino acid causing a crinkle in the peptide.

This manuscript has the potential to make a valuable contribution to the immunology of the domestic chicken and its many fascinating differences form the human counterpart.

The main weakness is the manuscript, which is too long, poorly structured and as a consequence impedes readers' comprehension.

---

## [Editor Report · Decision Letter 2]

31 Mar 2021

Dear Dr. Kaufman,

On behalf of my colleagues and the Academic Editor, Philippa Marrack, I am pleased to say that we can in principle offer to publish your Research Article "The dominantly-expressed class II molecule from a resistant MHC haplotype presents only a few Marek’s disease virus peptides by using an unprecedented binding motif" in PLOS Biology, provided you address any remaining formatting and reporting issues. Please ensure that the legends to all the applicable main and supplementary figures include a sentence to indicate where the underlying data can be found. E.g. "The underlying numerical data for this figure can be found in FILENAME. The rest of the formatting and reporting issues will be detailed in an email that will follow this letter and that you will usually receive within 2-3 business days, during which time no action is required from you. Please note that we will not be able to formally accept your manuscript and schedule it for publication until you have made the required changes.

PRESS

Thank you again for supporting Open Access publishing. We look forward to publishing your paper in PLOS Biology. 

Sincerely, 

Paula

---

Paula Jauregui, PhD 

Associate Editor 

PLOS Biology